# Impact of the SIGN head injury guidelines and NHS 4-hour emergency target on hospital admissions for head injury in Scotland: an interrupted times series

Carl Marincowitz,[1] Fiona E Lecky,[2] Eleanor Morris,[1] Victoria Allgar,[3] Trevor A Sheldon[4]

¹Hull York Medical School, Allam Medical Building, University of Hull, Hull, UK
²University of Sheffield, School of Health and Related Research, Sheffield, UK
³Hull York Medical School, John Hughlings, University of York, York, UK
⁴Department of Health Sciences, Alcuin Research Resource Centre, University of York, York, UK

**Correspondence to**
Dr Carl Marincowitz;
Carl.Marincowitz@hyms.ac.uk

## ABSTRACT

**Objectives** Head injury is a common reason for emergency department (ED) attendance. Around 1% of patients have life-threatening injuries, while 80% of patients are discharged. National guidelines (Scottish Intercollegiate Guidelines Network (SIGN)) were introduced in Scotland with the aim of achieving early identification of those with acute intracranial lesions yet safely reducing hospital admissions. This study aims to assess the impact of these guidelines and any effect the national 4-hour ED performance target had on hospital admissions for head injury.

**Setting** All Scottish hospitals between April 1998 and March 2016.

**Participants** Patients admitted to hospital for head injury or traumatic brain injury (TBI) diagnosed by CT imaging identified using administrative Scottish Information Services Division data. There are 275 hospitals in Scotland. In 2015/2016, there were 571 221 emergency hospital admissions in Scotland.

**Interventions** The SIGN head injury guidelines introduced in 2000 and 2009. The 4-hour ED target introduced in 2004.

**Outcomes** The monthly rate of hospital admissions for head injury and traumatic brain injury.

**Study design** An interrupted time series analysis.

**Results** The first guideline was associated with a reduction in monthly admissions of 0.14 (95% CI 0.09 to 4.83) per 100 000 population. The 4-hour target was associated with a monthly increase in admissions of 0.13 (95% CI 0.06 to 0.20) per 100 000 population. The second guideline reduced monthly admissions by 0.09 (95% CI–0.13 to −0.05) per 100 000 population. These effects varied between age groups. The guidelines were associated with increased admissions for patients with injuries identified by CT imaging—guideline 1: 0.06 (95% CI 0.004 to 0.12); guideline 2: 0.05 (95% CI 0.04 to 0.06) per 100 000 population.

**Conclusion** Increased CT imaging of head injured patients recommended by SIGN guidelines reduced hospital admissions. The 4-hour ED target and the increased identification of TBI by CT imaging acted to undermine this effect.

### Strengths and limitations of this study

► This is the first study to assess the impact of the SIGN head injury guidelines and 4-hour emergency department target on hospital admissions for head injury.
► We used the robust method of interrupted time series analysis and found the SIGN guidelines acted to reduce hospital admissions, but the 4-hour target increased hospital admissions.
► Due to the aggregated nature of the available data, we were unable to perform some age group-specific and injury subgroup sensitivity analysis.

## BACKGROUND

There are 1.4 million annual attendances to emergency departments (ED) in England and Wales following a head injury (blunt trauma to the head).[1] In Scotland, an estimated 6.6% of ED attendances are for head injury.[2] Approximately 95% of patients present with an initial Glasgow Coma Scale of 13–15, indicating normal or minimally impaired conscious level and are defined as having a minor head injury.[1 3]

Around 1% of minor head injured patients have life-threatening traumatic brain injuries (TBI) (injury to the brain/functional impairment due to external force), but this may not be initially clinically apparent.[1] Early identification of severe TBI can facilitate life-saving neurosurgery.[4] The clinical challenge is to differentiate patients with life-threatening TBI who present with a high conscious level from patients who can be discharged safely. This can be achieved through observation for deterioration or cranial CT imaging.[5] The health services challenge is finding a way of differentiating these groups in a way that minimises unnecessary imaging or

inpatient hospital admissions of patients without clinically important TBI. Research has focused on developing clinical decision rules that, using clinical assessment, triage patients at risk of life-threatening TBI to CT imaging and allow the discharge of low-risk patients. The most validated, in adults, is the Canadian CT Head Rule and this forms the basis of the Scottish Intercollegiate Guidelines Network (SIGN) and English National Institute for Health and Care Excellence (NICE) head injury guidelines.[1 2 6] The second SIGN guideline contains specific paediatric indications for CT imaging influenced by the CHALICE rule, which was derived in a population of head injured children.[2]

Two iterations of the SIGN guidelines have been introduced (figure 1).[2 7] The first recommended increased CT imaging of head injured patients but still featured a role for skull X-rays and admission for observation.[7] The second SIGN guideline extended CT imaging further.[2] A study assessing trends in hospital admissions for TBI in Scotland between 1998 and 2009 found changes in rates of admissions at specific time points after 2000, but the analysis was data driven and does not explicitly assess the impact of the SIGN guidelines.[8] Implementation studies of NICE head injury guidelines in England suggested that the cost of increased CT imaging (£100 per scan) would be offset by a reduction in inpatient hospital admissions (£847 per admission) as patients admitted for observation would be discharged from the ED following normal CT imaging, preventing many admissions.[9–11]

However, a study using English Hospital Episodes Statistics (HES) found head injury inpatient admissions increased following the introduction of the NICE guidelines, possibly due to the near simultaneous introduction of the 4-hour ED target. Online supplementary material 1 presents the details of the 4-target and in 2002 around 23% of all patients in the UK remained in the ED for longer than 4 hours.[12] The target could act to incentivise inpatient admissions for patients at medium risk for TBI, who require imaging within 8 hours, to avoid breaches. Previously, they would have been imaged in the ED and the majority discharged. A further cause of increased admissions could be the detection of brain injuries previously not identified, some without clinically significant sequelae, due to increased imaging.[13] In Scotland, the 4-hour target was not introduced at the same time as the SIGN guidelines. This provides a unique opportunity to assess the independent effects of these policy interventions.

## AIMS

This study aims to assess the impact of the introduction of the SIGN head injury guidelines and the 4-hour target on the rate of hospital admissions in patients with head injury and explores the extent to which any increase in admissions were due to the identification of more TBI.

## METHODS

### Study design

Interrupted time series analysis is an established and robust method for the evaluation of health policies implemented at discrete time points.[14 15]

### Dataset

The Scottish Information Services Division (ISD) contains a repository of information routinely collected at discharge from hospital for all non-obstetric and non-psychiatric inpatient hospital admissions in Scotland (including most short stay admissions to clinical decision units). Since 1996, reason for admission has been categorised using the International Statistical Classification of Diseases and Related Health Problems, 10th Revision (ICD10) diagnostic coding and since 1989 interventions have been coded using the OPSC4 classification.[16–18] This repository was used, for the period 1998–2016, to generate monthly numbers of patients admitted with ICD10 coding for head injury, patients admitted with ICD coding indicating TBI identified by CT imaging, neurosurgical interventions and deaths within 30 days of admission for patients with CT evidence of TBI. Diagnostic codes were counted if they were either primary or secondary diagnoses.

The data extract used for analysis is available from ISD Scotland for a commercial fee. The data set was fully anonymised and aggregated with small numbers suppressed.

We selected an ICD10 code subset definition of head injury that includes all possible definitions of TBI and head injury (table 1). The ICD10 codes used are consistent with those selected to explore the NICE guideline's effect on hospital admissions for head injury, with the addition of codes for crush injuries.[13] We selected an ICD10 code subset to define TBI that corresponds to injuries identified by CT imaging including codes for traumatic intracranial haemorrhages and skull fractures, but excluded codes for concussion and other clinical diagnoses (table 1).

### Outcomes

#### Admissions for head injury

The monthly number of patients admitted to hospital with one or more ICD10 code indicating head injury between April 1998 and March 2016 was generated by the ISD. As the SIGN guidelines have specific recommendations regarding paediatric patients and patients aged 65 years and over, admissions were stratified into: 0–15, 16–64 and ≥65 age groupings. Yearly and monthly number of admissions were converted into a rate per 100 000 population using Nomis Office of National Statistics (ONS) mid-year population estimates for Scotland for each age grouping (see online supplementary material 2).

#### Admissions for traumatic brain injury

The ISD generated the monthly number of patients admitted to hospital with at least one ICD10 code

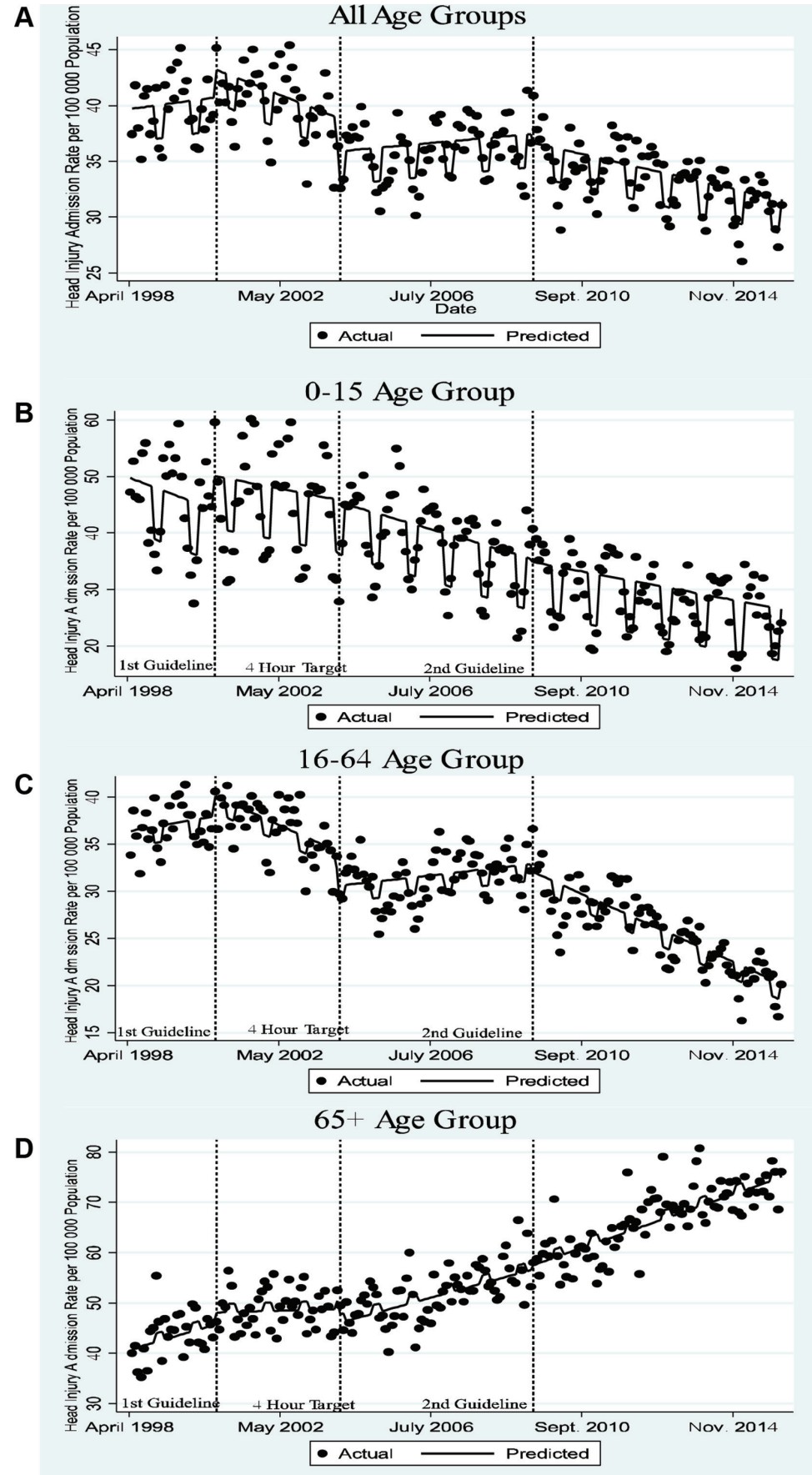

**Figure 1** The Impact of the 4-hour target and Scottish Intercollegiate Guidelines Network guidelines on head injury admissions.

**Table 1** Annual trends admissions for head injury and TBI

| Year | Annual rates head injury admissions per 100 000 population each age group* | | | | Annual rates TBI admissions per 100 000 population all age† (percentage annual rate all head injury admissions) |
|---|---|---|---|---|---|
| | All ages | 0–15 | 16–64 | 65+ | |
| 1999 | 484 | 558 | 453 | 536 | 77 (15.9%) |
| 2000 | 476 | 519 | 449 | 556 | 70 (14.7%) |
| 2001 | 493 | 561 | 457 | 578 | 72 (14.6%) |
| 2002 | 489 | 556 | 447 | 598 | 71 (14.5%) |
| 2003 | 451 | 521 | 404 | 581 | 76 (16.9%) |
| 2004 | 435 | 495 | 381 | 589 | 82 (18.9%) |
| 2005 | 417 | 497 | 350 | 591 | 78 (18.7%) |
| 2006 | 427 | 477 | 369 | 585 | 79 (18.5%) |
| 2007 | 442 | 441 | 393 | 653 | 79 (17.9%) |
| 2008 | 435 | 400 | 386 | 670 | 76 (17.4%) |
| 2009 | 440 | 406 | 377 | 709 | 79 (18%) |
| 2010 | 400 | 361 | 337 | 709 | 80 (20%) |
| 2011 | 421 | 369 | 352 | 741 | 91 (21.6%) |
| 2012 | 411 | 339 | 321 | 817 | 89 (21.7%) |
| 2013 | 396 | 307 | 293 | 845 | 97 (24.5%) |
| 2014 | 385 | 319 | 268 | 848 | 102 (26.5%) |
| 2015 | 373 | 316 | 251 | 867 | 123 (33%) |

*ICD10 codes for head injury: S00-S09, T04.0 and T06.0.
†ICD10 codes for TBI: S02.0, S02.1, S02.7, S02.8, S02.9, S06.1, S06.3, S06.4, S06.5, S06.6, S06.7, S06.8, S06.9, T04.0 and T06.0.
ICD, International Statistical Classification of Diseases and Related Health Problems, 10th Revision.

subclassification indicting an admission for TBI. Monthly and yearly rates per 100 000 population of admissions for TBI were calculated using Nomis ONS mid-year Scottish population estimates.

The ISD also provided the monthly number of patients admitted with an ICD10 code indicating TBI that had one or more OPSC4 neurosurgical codes (table 2). A monthly proportion of admissions for TBI that resulted in neurosurgery was estimated. The small number of monthly TBI admissions prevented release of data stratified by age group.

### Deaths related to traumatic brain injury
The ISD provided the monthly number of patients who died within 30 days of admission with at least one ICD10 code for TBI. This was converted into a rate using Nomis ONS mid-year Scottish population estimates. The monthly proportion of patients admitted with TBI who died within 30 days of admission was estimated using the total number of monthly admissions for ICD10 codes that corresponded to TBI.

### Statistical analysis
A monthly time series of the rate of inpatient hospital admissions for head injury ICD codes was plotted from April 1998 to March 2016. An interrupted times series analysis was completed assessing the impact of SIGN head injury guidelines and the 4-hour ED target using the ITSA package in STATA V.14.[19][20] The model included three intervention time points: the introduction of the first SIGN guideline in August 2000, the introduction of the 4-hour target in 2004 and the introduction of the second SIGN guideline in May 2009. Analysis was stratified into three age groups: 0–15, 16–64 and ≥65s. A segmented regression model predicting the rate of hospital admissions per 100 000 population in each age grouping per month was estimated.[15]

Autocorrelation of the residuals was assessed using the Durbin-Watson and Rho statistic.[15][21] Where there was sufficient deviation from a Durbin Watson statistic of 2 and the Rho statistic was not statistically significant, the Prais-Winsten transformation was used to adjust for autocorrelation.[15] Seasonality was assessed by introducing a dummy variable to the model in which winter months (December, January and February) were coded 1 and was included in the model when a statistically significant predictor.

The interrupted time series analysis was repeated to assess the impact of the SIGN guidelines and 4-hour target on the rate of hospital admissions for patients with an ICD10 code indicating TBI, the impact of SIGN guidelines on the proportion of inpatient admissions for TBI that resulted in neurosurgery or death and the death rate within 30 days of admission for TBI. It was thought a priori that the 4-hour target would not plausibly have affected neurosurgery or deaths. This was confirmed through visual inspection and was excluded from analysis of these outcomes.

**Table 2** Impact of the SIGN guidelines on number of admissions and deaths from TBI per 100 000 Scottish population

| Outcome | Winter effect | Initial trend | First SIGN guideline | 4-hour target introduced | Second SIGN guideline | Durbin-Watson statistic |
|---|---|---|---|---|---|---|
| Admissions for TBI/100 000 | –0.04 (95% CI –0.09 to 0.004), p=0.07 | | Change level: 0.26 (95% CI –0.74 to 1.26), p=0.61<br>Change trend: 0.06 (95% CI 0.004 to 0.12), p=0.04 | Change level: 0.16 (95% CI –0.67 to 0.99), p=0.71<br>Change trend: –0.02 (95% CI –0.05 to 0.01), p=0.24 | Change level: –0.39 (95% CI –1.09 to 0.30), p=0.27<br>Change trend: 0.05 (95% CI 0.03 to 0.07), p<0.01 | Untransformed 1.46 Prais-Winsten 2.02 |
| Percentage TBI admissions neurosurgical* | | 0.05 (95% CI –0.01 to 0.11), p=0.10 | Change level: –0.64 (95% CI –1.83 to 0.56), p=0.29<br>Change trend: –0.06 (95% CI –0.12 to 0.001), p=0.047 | | Change level: 0.47 (95% CI –0.17 to 1.12), p=0.15<br>Change trend: –0.01 (95% CI –0.03 to –0.003), p=0.01 | Untransformed 1.81 |
| Deaths/100 000 | 0.03 (95% CI 0.001 to 0.07), p=0.04 | –0.001 (95% CI –0.004 to 0.002), p=0.57 | Change level: –0.02 (95% CI –0.09 to 0.06), p=0.62<br>Change trend: 0.001 (95% CI –0.002 to 0.005), p=0.44 | | Change level: –0.01 (95% CI –0.06 to 0.05), p=0.85<br>Change trend: 0.0004 (95% CI –0.001 to 0.001), p=0.46 | Untransformed 2.3 |
| Percentage TBI admissions death | 0.88 (95% CI 0.36 to 1.41), p<0.01 | 0.03 (95% CI –0.03 to 0.10), p=0.35 | Change level: –0.79 (95% CI –2.12 to 0.54), p=0.24<br>Change trend: –0.03 (95% CI –0.10 to 0.03), p=0.33 | | Change level: 0.52 (95% CI –0.32 to 1.35), p=0.22<br>Change trend: –0.03 (95% CI –0.04 to –0.01), p<0.01 | Untransformed 2.08 |

*Neurosurgical procedure defined as one or more OPSC4 codes: A05.2, A05.3, A05.4, A05.8, A05.9, A40.1, A40.8, A40.9, A41.1, A41.8, A41.9, V03.1, V03.2, V03.4, V03.6, V03.7, V03.8, V03.9, V05.3 and V05.4.
SIGN, Scottish Intercollegiate Guidelines Network; TBI, traumatic brain injury.

Estimates of the impact of the policy interventions on inpatient admissions were made by using the preintervention model to estimate hypothetical monthly rates of admissions if no intervention had occurred. These were subtracted from the monthly admission rates estimated by the postintervention model.[14] To explore any effect of a policy implementation lag a sensitivity analysis was performed for all the models in which the 12 months immediately following the introduction of a policy change were excluded from analysis.[15]

### Patient and public involvement

The Hull and East Yorkshire NHS Trust Trans-Humber Consumer Research Panel and Hull branch of the Headway charity were consulted in the initial stages of developing the research questions addressed in this study. These patient groups highlighted that although national head injury guidelines seemed evidence based, there appeared to be little evidence to show they had achieved their aims.

## RESULTS

### Head injury inpatient hospital admissions

Table 1 show yearly rates of inpatient hospital admissions. Figure 1 and table 3 present the results of the interrupted time series assessing the impact of the SIGN guidelines and 4-hour ED target on monthly head injury admissions. Admission rates and estimates of effect are reported per 100 000 population in each age grouping. The SIGN guidelines and 4-hour target were associated with a significant change in the total level and trend of the rate admissions for head injury (figure 1A). The effect varied between age group.

### 0–15 age group

Monthly inpatient admissions fell from 47.18 to 24.02 per 100 000 per month over the time period (figure 1B). Neither SIGN guideline nor the introduction of the 4-hour target significantly affected the underlying reducing trend in hospital admissions but the first guideline was associated with a borderline statistically significant increase in level (6.06; 95% CI −0.49 to 12.62) (table 3). Admissions were less likely to occur in winter months (−9.19; 95% CI −10.81 to −7.57) (table 3).

### 16–64 age group

Inpatient admissions fell from 36.39 to 20.20 per 100 000 per month from April 1998 to March 2016 (figure 1C). Before the first SIGN guideline hospital admissions were increasing monthly (0.04; 95% CI −0.08 to 0.15) (table 3 and figure 1C). The first guideline was associated with a declining monthly trend in admissions (−0.20; 95% CI−0.35 to −0.05). The 4-hour target was associated with an initial fall in the number of inpatient admissions (−3.54; 95% CI −5.76 to −1.33), but subsequent trend of increasing monthly admissions (0.18; 95% CI 0.10 to 0.27). The second guideline was associated with a return to a declining trend in admissions (−0.18; 95% CI −0.23 to −0.13). Inpatient admissions were reduced in winter months (−1.74; 95% CI −2.48 to −1.01).

The trend following the introduction of the first SIGN guideline hypothetically continued in the period after the introduction of the 4-hour target is shown in online supplementary material 3. By subtracting this from the model that incorporated the introduction of the 4-hour target and the second SIGN guideline, we estimate that from January 2004 to March 2016 the introduction of the 4-hour target was associated with an additional 745 hospital admissions per 100 000 population aged 16–64 years.

### ≥65 age group

Monthly admissions increased from 40.00 to 76.09 over the time period (figure 1D). The only statistically significant change in the underlying trend was at the introduction of the 4-hour target which was associated with an acceleration in the increase in hospital admissions (0.15; 95% CI 0.01 to 0.28) (table 3 and figure 1D). Winter months were associated with increased hospital admissions (1.67; 95% CI 0.32 to 3.02).

### Sensitivity analysis for 12-month implementation lag

The introduction of an intervention lag in the model (online supplementary material 4) did not materially affect estimates of effect associated with interventions.

### Inpatient hospital admissions for traumatic brain injury on CT scan

Admission per 100 000 per month increased from 6.85 to 10.21 over the time period (figure 2). Before the first SIGN guideline hospital admissions were decreasing (−0.04; 95% CI −0.09 to 0.004) (table 2 and figure 2). The introduction of the first SIGN guideline was associated with a trend of increasing admissions (0.06; 95% CI 0.004 to 0.12). The 4-hour target was not associated with a significant change in level or trend. The introduction of the second SIGN guideline was associated with an acceleration in the increase in admissions (0.05; 95% CI 0.04 to 0.06).

Comparing admissions for TBI that would have occurred if the second SIGN guideline had not been introduced with the empirically derived model indicates that from May 2009 to March 2016 the introduction of the second SIGN guideline was associated with an additional 138 hospital admissions per 100 000 population (see online supplementary material 5).

Both SIGN guidelines were associated with a reduction in trend in the percentage of TBI admissions resulting in neurosurgery: first guideline (−0.06; 95% CI −0.12 to 0.001) and second guideline (−0.01; 95% CI −0.03 to −0.003) (table 2 and online supplementary material 6). A 12-month implementation lag did not materially affect the estimates (see online supplementary material 7).

**Table 3** Impact of the SIGN guidelines and introduction of 4-hour ED target on number of head injury admissions per 100 000 Scottish population by age group

| Age band | Winter effect | Initial trend | First SIGN guideline | 4-hour target introduced | Second SIGN guideline | Durbin-Watson statistic |
|---|---|---|---|---|---|---|
| All ages | −3.00 (95% CI −3.78 to −2.30), p<0.01 | 0.04 (95% CI −0.08 to 0.15), p=0.53 | Change level: 2.45 (95% CI 0.09 to 4.83), p=0.04<br>Change trend: −0.14 (95% CI −0.27 to −0.01), p=0.03 | Change level: −2.89 (95% CI −4.84 to −0.95), p<0.01<br>Change trend: 0.13 (95% CI 0.06 to 0.20), p<0.01 | Change level: −0.84 (95% CI −2.50 to 0.78), p=0.31<br>Change trend: −0.09 (95% CI −0.13 to −0.05), p<0.01 | Untransformed 1.68<br>Prais-Winsten 2.02 |
| 0–15 | −9.19 (95% CI −10.81 to −7.57), p<0.01 | −0.21 (95% CI −0.53 to 0.12), p=0.22 | Change level: 6.06 (95% CI −0.49 to 12.62), p=0.07<br>Change trend: −0.10 (95% CI −0.29 to 0.49), p=0.61 | Change level: −0.41 (95% CI −5.98 to 5.17), p=0.89<br>Change trend: −0.05 (95% CI −0.27 to 0.16), p=0.63 | Change level: −0.54 (95% CI −5.31 to 4.22), p=0.82<br>Change trend: 0.06 (95% CI −0.07 to 0.18), p=0.37 | Untransformed 1.34<br>Prais-Winsten 1.87 |
| 16–64 | −1.75 (95% CI −2.48 to −1.01), p<0.01 | 0.06 (95% CI −0.07 to 0.19), p=0.39 | Change level: 2.12 (95% CI −0.54 to 4.79), p=0.12<br>Change trend: −0.20 (95% CI −0.35 to −0.05), p<0.01 | Change level: −3.54 (95% CI −5.76 to −1.33), p<0.01<br>Change trend: 0.18 (95% CI −0.10 to 0.27), p<0.01 | Change level: −0.87 (95% CI −2.74 to 0.99), p=0.36<br>Change trend: −0.18 (95% CI −0.23 to −0.13), p<0.01 | Untransformed 1.45<br>Prais-Winsten 2.10 |
| 65+ | 1.67 (95% CI 0.32 to 3.02), p=0.02 | 0.17 (95% CI −0.04 to 0.39), p=0.11 | Change level: 2.28 (95% CI −2.13 to 6.86), p=0.30<br>Change trend: −0.16 (95% CI −0.40 to 0.09), p=0.21 | Change level: −2.70 (95% CI −6.39 to 0.99), p=0.15<br>Change trend: 0.15 (95% CI 0.01 to 0.28), p=0.03 | Change level: 0.85 (95% CI −2.22 to 3.93), p=0.59<br>Change trend: 0.05 (95% CI −0.03 to 0.12), p=0.23 | Untransformed 1.70<br>Prais-Winsten 2.00 |

ED, emergency department; SIGN, Scottish Intercollegiate Guidelines Network; TBI, traumatic brain injury.

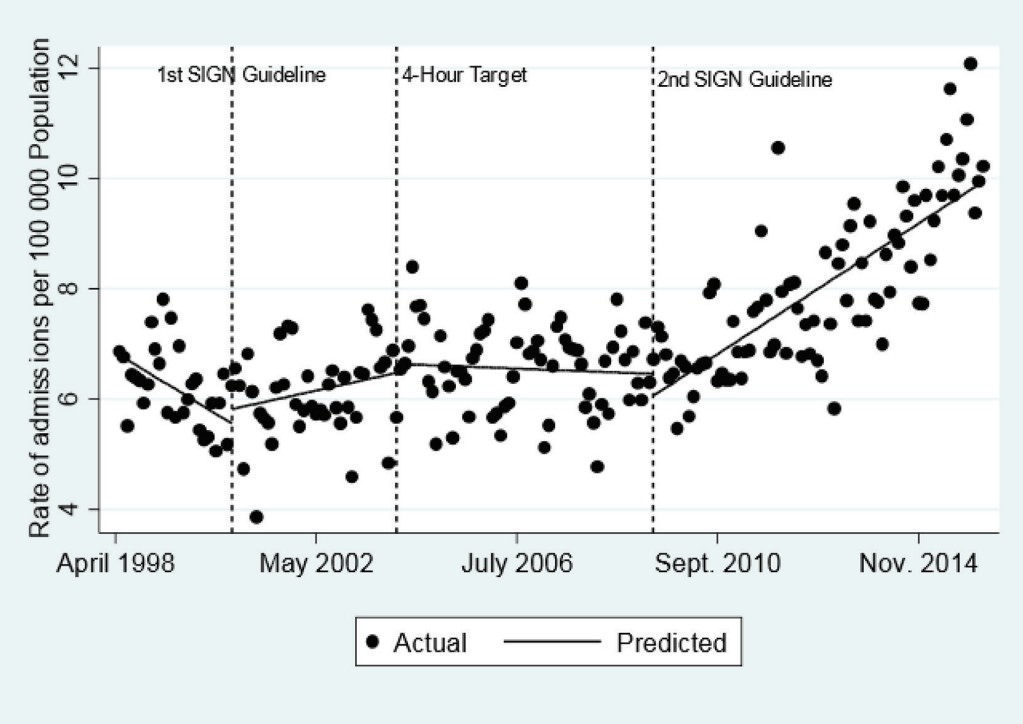

**Figure 2** The impact of the Scottish Intercollegiate Guidelines Network (SIGN) guidelines and 4-hour target on admissions for traumatic brain injury.

### Deaths following admission for TBI

Neither guideline was associated with a change in level or trend in monthly death rate within 30 days of admission with an ICD10 code indicating TBI (see table 2 and online supplementary material 8). The introduction of the second SIGN guideline was associated with a significant reduction in the underlying trend in the monthly percentage of inpatient admissions for TBI that resulted in death within 30 days (−0.03; 95% CI −0.04 to −0.01) (see table 2 and online supplementary material 9).

Introduction of a 12-month implementation lag into the models did not materially change the estimates (see online supplementary material 10).

## DISCUSSION
### Summary

To our knowledge, this is the first study to evaluate the impact of the SIGN guidelines and the 4-hour ED target on head injury hospital admissions. We found evidence that the SIGN guidelines reduced inpatient admissions, the effect varying by age group. In the 16–64 age group, both guidelines were associated with a reduction in hospital admissions (figure 1C and table 3). This may be due to increased CT imaging in the ED identifying more patients without TBI safe for discharge. In the paediatric population, there was an underlying trend of reducing hospital admissions that the guidelines did not appear to affect. In the ≥65 age group, neither SIGN guideline iteration acted to offset the secular trend of increasing hospital admissions. Inpatient admissions increased in

winter months in the ≥65 age group but were reduced in other age groups. Falls from standing are known to be the most common cause of brain injuries in those aged ≥65 years, while assault and road traffic accidents are more common causes of injuries in younger patients.[22] Weather conditions during winter may increase the likelihood of falls from standing while reducing the prevalence of assault or road traffic accidents.[23]

The introduction of both SIGN guideline iterations were associated with an increased in hospital admissions for patients with TBI and a reduction in the proportion of inpatient admissions that resulted in neurosurgery or death (table 2). The guidelines may have acted, as previously hypothesised, to increase CT diagnosis and admissions of patients with brain injuries of lower severity who do not require intervention.[13] The 4-hour target was associated with an increase in hospital admissions for adults (figure 1C and D and table 3). This effect was reversed by the second SIGN guideline in the 16–64 but not the ≥65 age group.

The 4-hour target's effect on adult head injury hospital admissions appears related to the time from admission which CT imaging is recommended. The first guideline contained no recommendation for when CT imaging should occur. The introduction of the 4-hour ED target increased head injury hospital admissions in both the 16–64 and ≥65 age groupings, presumably as patients were admitted to await imaging to comply with the target. The second SIGN guideline increased the indications for immediate CT imaging in the 16–64 age group and

was associated with a downward trend in admissions. A reduction in hospital admissions was not observed in the ≥65 age group following the second SIGN guideline. The second SIGN guideline includes a series of specific additional indications for imaging in the ≥65 age group that are recommended to occur within 8 hours of ED attendance. As 8 hours is longer than the 4-hour target patients with these indications for CT imaging would be admitted to hospital. This may account for why the second SIGN guideline was not associated with reduced hospital admissions for those aged ≥65 years.

## Strengths

We have used a time series of 216 data points and followed established techniques to control for seasonal factors and autocorrelation.[15] We have adjusted for population factors using mid-year population estimates that incorporate the changing demography of Scotland's population. The models constructed were robust to sensitivity analysis for time lags.

There is controversy regarding which ICD10 codes correspond to clinical definitions of head injury and TBI with inconsistent sets of ICD10 codes used to encompass both.[24] We selected subsets of codes for head injury that are likely to be sensitive to changes in admission practice related to increased diagnostic precision of TBI from increased CT imaging and inpatient admissions of patients awaiting an exclusion of TBI due to the 4-hour target. Our ICD10 code selection for TBI was intended to encompass radiologically detected injuries as this outcome would be sensitive to increased diagnoses of TBI by CT imaging.

## Weaknesses

Ideally, the effects of guideline implementation and policy interventions would be assessed using randomised control trials. However, interrupted time series analysis (a rigorous quasi-experimental study design) is becoming increasingly popular particularly for the evaluation of healthcare practice, programmes and policy because it allows causal inferences when interventions are introduced at specific time points. Discontinuities in outcomes, observed at or shortly after the time of intervention, constitute persuasive evidence of an effect with high internal and external validity.[14 15 25 26] The method has limitations notably the potential for observed discontinuities to result from co-interventions instead of the health policies under investigation. We cannot find other policies or sudden changes to the population of Scotland that could account for the observed changes in admissions for head injury in Scotland at the time of the either the introduction of the SIGN guidelines or the 4-hour ED target.

This study used routinely collected Scottish ISD data, and administrative data should be approached with some caution.[27] There may be inaccuracies in diagnostic codes due to coding errors. However, there were no changes to the cohort of admitted patients that data were collected

on during the study period and ISD data have been found to be both sufficiently reliably and comprehensively collected to support its use in research.[28 29] Furthermore, random poor coding, without changes in coding practice, are unlikely to result in sharp discontinuities at the specific time points of the policies considered here. ICD coding changed in 1996, so we only used ISD data from 1998 to give time for adjustment. This limited the number of data points before the first SIGN guideline, so we may have lacked the power to detect some changes associated with the first SIGN guideline as statistically significant (table 3 and figure 1).

We could not stratify all the analysis by age group as the small number of some outcomes prevented release of aggregated data. A sensitivity analysis using ICD10 injury subtypes was also not possible due to the aggregated nature of the available data. A more sensitive outcome measure for changes in admission practice due to either the SIGN guidelines or 4-hour target may be the proportion of attendances to the ED following head injury that result in inpatient hospital admission. TBI has become more common in the elderly and if analysis of the proportion of ED attendances that resulted in inpatient admission was stratified by age this would account for age group differences in incidence of injuries.[30] We were unable to differentiate clinical decision unit admissions from other types of inpatient admissions in our data. However, the extent to which clinical decision unit admissions in the UK represent materially different and more cost-effective care compared with other types of hospital admissions and should be treated differently is debatable.[31] Furthermore, only six hospitals in Scotland had clinical decision units during the time period of our study and were not established at the same time as the policies considered here.[32]

There is evidence that deaths from severe TBI fell following the introduction of the NICE guidelines.[33] We may have missed this effect due to the undifferentiated cohort of patients with TBI used to assess deaths and the use of all-cause mortality. We also could not adjust mortality by age or injury severity.

Estimates of the impact of guidelines will depend on the extent to which the guidelines have been implemented. There are no national audit data on this, although one local audit conducted in 2001 indicated less than half of patients the SIGN guidelines deemed were safe for discharge from the ED were actually discharged.[34] Our study may therefore underestimate the potential impact of full implementation.

## Comparison to previous literature

A study that assessed the NICE head injury guideline's impact using English data (although not using an interrupted time series) found that increased CT imaging led to an increase in hospital admissions, contrary to expectations.[13] Increased admissions following the introduction of the NICE guidelines could be due to increased diagnosis of TBI due to more CT imaging or the effect of the

4-hour ED target.[13] We found evidence that both factors increased hospital admissions in Scotland.

Analysis of English HES data from 2000 to 2011 found yearly paediatric hospital admissions for head injury increased from 34 to 37 inpatient admissions per 10 000 children.[35] We found Scottish paediatric admissions fell from 56 per 10 000 children in 1999 to 32 per 10 000 children in 2015. There is evidence that clinicians are less likely to implement head injury guidelines in children.[35–37] A lack of implementation may explain why the SIGN guidelines were not associated with a reduction in paediatric hospital admissions.

## Implications

The overall effect of the SIGN guidelines was to reduce inpatient hospital admissions for head injured patients and this supports previous research indicating early CT imaging may represent a cost-effective management strategy.[9] However, the guidelines were associated with increased inpatient admissions for patients with TBI, possibly resulting from increased imaging identifying more TBI of lower clinical severity. Research better characterising the risk associated with TBI identified by CT imaging could help identify a subset of low-risk patients who could be safely discharged from the ED. This could help mitigate the increase in inpatient hospital admissions in this group associated with the SIGN and other similar guidelines.

We found evidence that the 4-hour ED target acted to reverse reductions in hospital admissions in the 16–64 age group associated with the SIGN guidelines. As has been previously argued, performance targets need to be carefully considered before implementation to ensure that they do not have unintended consequences, in this case undermining the benefits of evidence-based clinical guidelines.[38] A more granular approach to the 4-hour ED target that reflects condition-specific clinical circumstances, such as the time frame of CT imaging in head injury, could help to prevent such costly unintended consequences.

Given the limitations in the mortality analysis undertaken, it is hard to draw conclusions about how effective the SIGN guidelines were at reducing deaths from TBI. Future analysis should attempt to adjust for age and severity of injury and this will require patient-level data.

## CONCLUSION

Increased early CT imaging of head injured patients may reduce hospital admissions. However, this effect may be offset by an increase in the diagnosis of TBI of lower severity and the 4-hour ED target. Future research should aim to better risk stratify patients with TBI identified by CT imaging to help reduce hospital admissions related to increased CT imaging. Care should also be taken when introducing arbitrary performance targets, such as the 4-hour target, to ensure they do not undermine the beneficial effect of clinical guidelines.

**Acknowledgements** Sir Graham Teasdale commented on the manuscript and provided invaluable historical context and knowledge regarding the use of administrative data sets in traumatic brain injury research. The Hull and East Yorkshire NHS Trust Trans-Humber Consumer Research Panel and Hull branch of the Headway charity helped develop the research questions addressed in this study.

**Contributors** This idea for the study was conceived by CM with help from TS, FEL and VA. The analysis was completed by CM and EM with specialist advice regarding interrupted time series analysis from TS and VA. FL provided specialist advice regarding the clinical context and interpretation of the results. All authors read and approved the final manuscript.

**Funding** Carl Marincowitz is funded by a National Institute for Health Research Doctoral Fellowship (DRF-2016-09-086). This study presents independent research funded by the National Institute for Health Research (NIHR).

**Disclaimer** The views expressed are those of the author(s) and not necessarily those of the NHS, the NIHR or the Department of Health.

**Competing interests** None declared.

**Patient consent** Not required.

**Provenance and peer review** Not commissioned; externally peer reviewed.

**Data sharing statement** The data used for this analysis was purchased for a fee from Information Service Division Scotland. The corresponding author is happy to share the exact details of the data extract and how to acquire it from ISD Scotland.

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
