## [Reviewer comments · BMJ Open]

ARTICLE DETAILS

TITLE (PROVISIONAL)	The impact of the SIGN head injury guidelines and NHS 4-hour Emergency Target on hospital admissions for head injury in Scotland: An Interrupted Times Series.
AUTHORS	Marincowitz, Carl; Lecky, Fiona; Morris, Eleanor; Allgar, Victoria; Sheldon, Trevor

VERSION 1 – REVIEW

REVIEWER	Isaac Kobe University Hospitals of North Midlands, England
REVIEW RETURNED	02-May-2018

GENERAL COMMENTS	This is an interesting paper as most recent literature on this subject had been focused on effects of reducing imaging rather than looking at admission rates. A running head is usually required before the abstract. The abstract could do with more specifics and figures eg A small number of patients have life threatening injuries (PERCENTAGE?) whilst the majority (PERCENTAGE?) are discharged. All Scottish hospitals (HOW MANY ARE THESE?) Participants admitted to hospital. (THIS IS YOUR SAMPLE SIZE, SO HOW MANY?) It is not clear why some words are put in quotation marks eg minor/breaches. These are proper English words. Furthermore minor and mild head injury have different definitions. They are used interchangeably in the text suggesting they have the same definition. I am not sure if Caanadian CT head rule is the most validated clinical decision rules (CDR) on head CT scans for all ages. There are several CDRs and in children up to 18 years, PECARN head CT rule has been found to be most validated sensitive CDR (metanalysis level evidence). A justification for reducing admission rates which is the hypothesis being tested hasn't been well done. This might be either a cost benefit analysis as well as other advantages of reduction of unnecessary admissions. The tables are well illustrated and the statistical analysis is sound. Limitations of the study are well noted by the authors. It would be interesting to assess compounded overall effect of SIGN guidelines and as well 4 hr rule on admission rates but given these were introduced at different times and such an effect would be difficult to assess. Overall this is a well conducted study with sound analysis.
--

REVIEWER	John DeAngelis MD, RDMS Cambridge Health Alliance, USA
REVIEW RETURNED	10-May-2018

GENERAL COMMENTS	While retrospective review and time analysis can be fraught with methodological issues, it appears that the authors have gone to great lengths to address these issues including seasonality and the often overlooked issue of using ICD codes for selection. While it is difficult to draw conclusions from retrospective data the authors claims are factual and without significant overreach. I think this paper well describes the changes put into place during periods of evidence based medicine and perhaps less evidenced based policy.
---

REVIEWER	Brad Wright University of Iowa, USA
REVIEW RETURNED	14-Jun-2018

GENERAL COMMENTS	This manuscript reports on a study to examine the impact of two policies (SIGN protocol and the 4-hour target) on hospital admissions for TBI in Scotland. The study is rather interesting and I have only a few comments. First, it is unclear how placement on a clinical decision unit factors in here. My understanding is that CDU-placement stops the 4-hour clock, but are you including such short stays as "admissions" or not? It would make sense to me that there would be an uptick in CDU visits as they would stop the clock to avoid 4-hour breaches, while also affording the amount of time needed to perform a rule-out head CT. Please clarify how this was handled and comment on it in the paper as a strategy for ensuring patient safety while complying with both policies. I found some parts of the discussion a bit rambling. Please consider a bit of a revision to better organize the presentation of that material. It just felt as if it jumped around a bit (apologies that that comment is itself rather vague). I would also recommend toning down your language around interrupted time series designs just a bit. I think this a perfectly reasonable method to employ here, but it does make some significant assumptions that you have very little way of validating other than to say "we aren't aware of any co-occurring policies, etc. etc." Arguably, a difference-in-differences model is superior, given its use of a control group, but even it is subject to assumptions (e.g., parallel trends) that often fail. All of this is to say, you've selected the right method, but you'd do best to frankly acknowledge its limitations, rather than attempt to oversell them. Otherwise, well done and important work.
---

VERSION 1 – AUTHOR RESPONSE

Reviewer 1:

A running head is usually required before the abstract.

This journal has been quite specific with the instructions regarding the presentation of the abstract and they did not indicate they wanted or required a running head.

The abstract could do with more specifics and figures eg A small number of patients have life threatening injuries (PERCENTAGE?) whilst the majority (PERCENTAGE?) are discharged. All Scottish hospitals (HOW MANY ARE THESE?) Participants admitted to hospital. (THIS IS YOUR SAMPLE SIZE, SO HOW MANY?)

We have amended the abstract to include specific estimates of the percentage of patients who have life-threatening injuries and who are discharged. We have added an ISD estimate for the annual number of patients admitted as an emergency to hospitals in Scotland in 2015/2016 for context and the number of hospitals.

All hospitals in Scotland over the time-period of analysis had to submit the number of acute admissions with ICD 10 diagnostic coding to ISD Scotland in order to receive payment in the NHS internal market for the treatment of patients. This yearly and monthly number was used to calculate the reported rates.

It is not clear why some words are put in quotation marks eg minor/breaches. These are proper English words.

Quotation marks have been removed.

Furthermore minor and mild head injury have different definitions. They are used interchangeably in the text suggesting they have the same definition.

We agree that mild and minor head injury define head injured populations with a different initial GCS score.

The 3rd sentence of the 1st paragraph of the section entitled background (page 4) has been changed to prevent any confusion. The only other time the word mild is used is in the title of a reference.

I am not sure if Canadian CT head rule is the most validated clinical decision rules (CDR) on head CT scans for all ages. There are several CDRs and in children up to 18 years, PECARN head CT rule has been found to be most validated sensitive CDR (metanalysis level evidence).

We agree that the Canadian CT head rule has predominantly been validated in adults and that other clinical decision rules have higher level evidence supporting their use in children.

The final sentence of the second paragraph of the background (page 4) has been altered to highlight this distinction. Early iterations of both the SIGN and NICE head injury guidelines were solely based on the Canadian CT head Rule as specific paediatric decision rules had not been derived or validated at the time. The 2nd SIGN guideline contains specific paediatric indications for CT imaging based on the CHALICE rule. We have added the final sentence of the second paragraph of the background (page 4) to highlight this.

A justification for reducing admission rates which is the hypothesis being tested hasn't been well done. This might be either a cost benefit analysis as well as other advantages of reduction of unnecessary admissions.

The final sentence of the 3rd paragraph of the background section (pages 4) now quantifies the extra cost of admitting a patient for observation compared to a CT scan.

It would be interesting to assess compounded overall effect of SIGN guidelines and as well 4 hr rule on admission rates but given these were introduced at different times and such an effect would be difficult to assess.

We agree it would be interesting to attempt to estimate this, but as the reviewer says, it would be difficult given the nature of the available data.

Reviewer 2:

While retrospective review and time analysis can be fraught with methodological issues, it appears that the authors have gone to great lengths to address these issues including seasonality and the often overlooked issue of using ICD codes for selection. While it is difficult to draw conclusions from retrospective data the authors claims are factual and without significant overreach. I think this paper well describes the changes put into place during periods of evidence based medicine and perhaps less evidenced based policy.

Thank you for your kind comments.

Reviewer 3:

First, it is unclear how placement on a clinical decision unit factors in here. My understanding is that CDU-placement stops the 4-hour clock, but are you including such short stays as "admissions" or not? It would make sense to me that there would be an uptick in CDU visits as they would stop the clock to avoid 4-hour breaches, while also affording the amount of time needed to perform a rule-out head CT. Please clarify how this was handled and comment on it in the paper as a strategy for ensuring patient safety while complying with both policies.

We have contacted Information Service Division Scotland to clarify how admissions to Clinical Decision Units were counted in our data set. ISD Scotland stated that the vast majority of Health Boards in Scotland counted and submitted CDU admissions as they did all other emergency inpatient admissions over our study time period. Therefore, CDU admissions form part of the estimated monthly rate of inpatient admissions in our time series analysis. The 1st sentence of paragraph of the section entitled data set has been amended to highlight this (page 5).

ISD Scotland also confirmed that no Scottish Health Board changed how they submitted CDU data over the time-period of our study.

The reviewer raises an interesting research question about the potential role of CDUs. However, the data available to us is an aggregated count of inpatient hospital admissions, so we cannot address it in this study.

I found some parts of the discussion a bit rambling. Please consider a bit of a revision to better organize the presentation of that material. It just felt as if it jumped around a bit (apologies that that comment is itself rather vague).

We have made the following changes to improve the presentation of the discussion. The second paragraph of the Summary section (page 15) has been moved to the Strengths section (page 15). The second paragraph of the Comparison to Previous Literature section (page 17) has been moved to the Summary section (page 15).

I would also recommend toning down your language around interrupted time series designs just a bit. I think this a perfectly reasonable method to employ here, but it does make some significant

assumptions that you have very little way of validating other than to say "we aren't aware of any co-occurring policies, etc. etc." Arguably, a difference-in-differences model is superior, given its use of a control group, but even it is subject to assumptions (e.g., parallel trends) that often fail. All of this is to say, you've selected the right method, but you'd do best to frankly acknowledge its limitations, rather than attempt to oversell them.

The language in the first paragraph of the Weaknesses section and Comparison to Previous Literature section has been changed to tone down the language around interrupted time series design.

VERSION 2 – REVIEW

REVIEWER	Isaac Kobe University Hospitals of North Midlands, Stoke on Trent, England
REVIEW RETURNED	31-Jul-2018

GENERAL COMMENTS	Good work with the corrections. Very well presented graphical statistics! Still a few things needing tidy up:-  -Pg 4 line 4:(Background) space between ' minimally' and the next word -last paragraph pg 6 different line spacing from the rest of the document. Otherwise a brilliant paper!
---

REVIEWER	Brad Wright University of Iowa, USA
REVIEW RETURNED	26-Jul-2018

GENERAL COMMENTS	As I said before, this is an interesting and important topic and the authors have done a decent job of responding to the prior reviews. However, I remain concerned that they lump clinical decision units in as admissions. I think that it is important to look at the CDU, but that these should be modeled separately from typical inpatient admissions, because the length of stay in CDU is very short (typically no more than 12 hours) -- yet moving a patient to CDU stops the clock with respect to the 4-hour target. So CDU can be used to game this rule. But that will show up very differently than longer inpatient admissions to a medical assessment unit or other ward. I strongly urge the authors to perform this stratified analysis.
--

VERSION 2 – AUTHOR RESPONSE

Reviewer: 1

Reviewer Name: Isaac Kobe

Institution and Country: University Hospitals of North Midlands, Stoke on Trent, England

Please state any competing interests or state 'None declared': None declared

Good work with the corrections. Very well presented graphical statistics!

Still a few things needing tidy up:-

-Pg 4 line 4:(Background) space between ' minimally' and the next word

-last paragraph pg 6 different line spacing from the rest of the document.

Otherwise a brilliant paper!

Our Response:

Thank you for your kind comments.

We have made the formatting changes suggested and these are highlighted using track changes.

Reviewer: 3

Reviewer Name: Brad Wright

Institution and Country: University of Iowa, USA

Please state any competing interests or state 'None declared': None declared

As I said before, this is an interesting and important topic and the authors have done a decent job of responding to the prior reviews. However, I remain concerned that they lump clinical decision units in as admissions. I think that it is important to look at the CDU, but that these should be modeled separately from typical inpatient admissions, because the length of stay in CDU is very short (typically no more than 12 hours) -- yet moving a patient to CDU stops the clock with respect to the 4-hour target. So CDU can be used to game this rule. But that will show up very differently than longer inpatient admissions to a medical assessment unit or other ward. I strongly urge the authors to perform this stratified analysis.

Our Response:

Thank you for the time you have taken in reviewing our paper and your useful comments. We agree that assessing whether the role CDUs played in the UK changed in response to the introduction of the 4-hour target and how they can be used in Emergency Department management of head injury are interesting research topics. However, as explained below the suggested analysis is not possible from the data provided by Information Services Division Scotland. We also think that CDU use has a limited effect on our analysis due to the small number of CDUs in Scotland, their role and the type of

study design we have used and so the lack of the analysis you suggest does not significantly weaken our research.

Admissions data is provided by the Information Services Division Scotland (ISD) and is the aggregated monthly number of inpatient hospital admissions in Scotland with ICD10 codes related to head injury and traumatic brain injury. It is not possible in these data to distinguish between admissions to CDUs and other types of inpatient hospital admissions. In response to your comments, we contacted the Information Service Division Scotland and they confirmed that even with individual level data it is not possible to distinguish CDU from other types of hospital admissions. We are unaware of any national administrative dataset for Scotland that has the granularity necessary to undertake the analysis that is being requested.

In Scotland, only 6 hospitals had Emergency Department CDUs out of a total of 36 Emergency Departments.[1 2] In addition, these CDUs have selective admission criteria and not all 6 admit patients with head injuries.[3] Consequently, only a small number of admissions for head injury in Scotland will be to CDUs, representing a small proportion of the head injury admissions in Scotland from which we have derived our analysis and drawn our conclusions.

The model of Clinical Decision Units/Short Stay Wards in the UK is often an inpatient admission to an acute ward such as a medical assessment unit under the care of an Emergency Medicine Consultant and there is limited evidence that the use of CDUs in the UK, unlike the USA, has been cost saving.[4 5] Therefore, we believe a hospital admission to a CDU whether to avoid breaching the 4-hour target and/or to comply with CT imaging guidelines in head injury does not constitute gaming. The care received by such patients admitted to CDUs is materially more intensive than that received in the rest of the Emergency Department. Although CDU admissions may be shorter than some other inpatient admissions, they represent a use of health care resources greater than management solely in the Emergency Department and like other forms of inpatient hospital admissions for short periods of time. We therefore don't think that CDU admissions should necessarily be regarded differently from other types of short inpatient admissions such as to Medical and Surgical Assessment Units. The aim of the study is to look at whether the SIGN guidelines and the 4 hour target resulted in changes in admissions (and so resource use and inconvenience for patients). If some of the increase in admissions we observe is due to more patients being admitted to a CDU this is still important.

The interrupted time series design means we are testing for discontinuities in trend and slope of admissions around the time point of policy interventions (SIGN guidance and 4 hour target). In 2010 a small minority of hospitals in Scotland had CDUs. CDUs were not introduced or their number expanded particularly in Scotland at the time that these policy interventions were introduced. We therefore don't think that CDU use in Scotland affects our analysis or substantive findings.

In order to recognise the issues you raise, however, we have revised the 3rd paragraph of the weaknesses section of the discussion to explain the role CDUs had in Scotland during the time period of our study and why it is not possible to conduct a sub-group analysis of CDU admissions .

1. www.audit-scotland.gov.uk/docs/health/2010/nr_100812_emergency_departments.rtf
2. <http://www.isdscotland.org/Health-Topics/Emergency-Care/Emergency-DepartmentActivity/Hospital-Site-List/>
3. www.gov.scot/Publications/2007/10/23093529/25
4. Goodacre SW. Role of the short stay observation ward in accident and emergency departments in the United Kingdom. *J Accid Emerg Med* 1998;15(1):26-30. [published Online First: 1998/02/25]
5. Cooke MW, Higgins J, Kidd P. Use of emergency observation and assessment wards: a systematic literature review. *Emerg Med J* 2003;20(2):138-42. [published Online First: 2003/03/19]

VERSION 3 – REVIEW

REVIEWER	Brad Wright University of Iowa, USA
REVIEW RETURNED	28-Sep-2018
GENERAL COMMENTS	The authors have now adequately addressed all concerns.